# High PrEP uptake and objective longitudinal adherence among HIV-exposed women with personal or partner plans for pregnancy in rural Uganda: A cohort study

Lynn T. Matthews[1]*, Esther C. Atukunda[2], Moran Owembabazi[2], Kato Paul Kalyebera[2,3], Christina Psaros[4,5], Pooja Chitneni[5,6], Craig W. Hendrix[7], Mark A. Marzinke[7,8], Peter L. Anderson[9], Oluwaseyi O. Isehunwa[1], Kathleen E. Hurwitz[10], Kara Bennett[11], Winnie Muyindike[2,3], David R. Bangsberg[12], Jessica E. Haberer[5,13], Jeanne M. Marrazzo[1‡], Mwebesa Bosco Bwana[3†‡]

1 Division of Infectious Diseases, University of Alabama at Birmingham, Birmingham, Alabama, United States of America, 2 Mbarara University of Science and Technology, Mbarara, Uganda, 3 Mbarara Regional Referral Hospital, Mbarara, Uganda, 4 Behavioral Medicine Program, Department of Psychiatry, Massachusetts General Hospital, Massachusetts, United States of America, 5 Harvard Medical School, Boston, Massachusetts, United States of America, 6 Division of Infectious Diseases and General Internal Medicine, Brigham and Women's Hospital, Boston, Massachusetts, United States of America, 7 Department of Medicine, Johns Hopkins University School of Medicine, Baltimore, Maryland, United States of America, 8 Department of Pathology, Johns Hopkins University School of Medicine, Baltimore, Maryland, United States of America, 9 Department of Pharmaceutical Sciences, Skaggs School of Pharmacy and Pharmaceutical Sciences, University of Colorado, Aurora, Colorado, United States of America, 10 NoviSci, Inc., a Target RWE company, Durham, North Carolina, United States of America, 11 Bennett Statistical Consulting Inc., Ballston Lake, New York, United States of America, 12 School of Public Health, Oregon Health Sciences University–Portland State University, Portland, Oregon, United States of America, 13 Center for Global Health, Massachusetts General Hospital, Boston, Boston, Massachusetts, United States of America

† Deceased.
‡ These authors are co-senior authors on this work.
* lynnmatthews@uabmc.edu

## Abstract

### Background

In Uganda, fertility rates and adult HIV prevalence are high, and many women conceive with partners living with HIV. Pre-exposure prophylaxis (PrEP) reduces HIV acquisition for women and, therefore, infants. We developed the Healthy Families-PrEP intervention to support PrEP use as part of HIV prevention during periconception and pregnancy periods. We conducted a longitudinal cohort study to evaluate oral PrEP use among women participating in the intervention.

### Methods and findings

We enrolled HIV–negative women with plans for pregnancy with a partner living, or thought to be living, with HIV (2017 to 2020) to evaluate PrEP use among women participating in the Healthy Families-PrEP intervention. Quarterly study visits through 9 months included HIV and pregnancy testing and HIV prevention counseling. PrEP was provided

**Data Availability Statement:** The full dataset is available in the Harvard Dataverse at https://doi.org/10.7910/DVN/4U4TTL.

**Funding:** This work was supported by the Doris Duke Clinician Scientist Development Award (LTM) and AMC21 at the University of Alabama at Birmingham (LTM). Gilead Sciences provided study drug and reviewed the final manuscript. These entities had no role in study design, data collection and analysis, decision to publish, or preparation of the manuscript.

**Competing interests:** LTM has received clinical research funding from Gilead Sciences. CWH has received clinical research funding from Gilead Sciences and Merck and, holds patents related to HIV prevention products licensed by Johns Hopkins to Prionde Biopharma, LLC, which he founded. PLA has received consultant fees from Merck, ViiV, and Gilead and research support from Gilead. JEH has received consultant fees from Merck and Natera. KEH is an employee and a shareholder of NoviSci/Target RWE, which has received fees from Janssen Research & Development (Janssen R&D) and Amgen Inc. outside the submitted work.

**Abbreviations:** ART, antiretroviral therapy; DBS, dried blood spot; GEE, generalized estimating equation; MoH, Ministry of Health; PK, pharmacokinetic; PrEP, pre-exposure prophylaxis; STI, sexually transmitted infection; TDF, tenofovir disoproxil fumarate; TFV, tenofovir; TFV-DP, TFV-diphosphate.

in electronic pillboxes, providing the primary adherence measure ("high" adherence when pillbox was opened $\geq$80% of days). Enrollment questionnaires assessed factors associated with PrEP use. Plasma tenofovir (TFV) and intraerythrocytic TFV-diphosphate (TFV-DP) concentrations were determined quarterly for women who acquired HIV and a randomly selected subset of those who did not; concentrations TFV $\geq$40 ng/mL and TFV-DP $\geq$600 fmol/punch were categorized as "high." Women who became pregnant were initially exited from the cohort by design; from March 2019, women with incident pregnancy remained in the study with quarterly follow-up until pregnancy outcome. Primary outcomes included (1) PrEP uptake (proportion who initiated PrEP); and (2) PrEP adherence (proportion of days with pillbox openings during the first 3 months following PrEP initiation). We used univariable and multivariable-adjusted linear regression to evaluate baseline predictors selected based on our conceptual framework of mean adherence over 3 months. We also assessed mean monthly adherence over 9 months of follow-up and during pregnancy.

We enrolled 131 women with mean age 28.7 years (95% CI: 27.8 to 29.5). Ninety-seven (74%) reported a partner with HIV and 79 (60%) reported condomless sex. Most women ($N$ = 118; 90%) initiated PrEP. Mean electronic adherence during the 3 months following initiation was 87% (95% CI: 83%, 90%). No covariates were associated with 3-month pill-taking behavior. Concentrations of plasma TFV and TFV-DP were high among 66% and 47%, 56% and 41%, and 45% and 45% at months 3, 6, and 9, respectively. We observed 53 pregnancies among 131 women (1-year cumulative incidence 53% [95% CI: 43%, 62%]) and 1 HIV-seroconversion in a non-pregnant woman. Mean pillcap adherence for PrEP users with pregnancy follow-up ($N$ = 17) was 98% (95% CI: 97%, 99%). Study design limitations include lack of a control group.

## Conclusions

Women in Uganda with PrEP indications and planning for pregnancy chose to use PrEP. By electronic pillcap, most were able to sustain high adherence to daily oral PrEP prior to and during pregnancy. Differences in adherence measures highlight challenges with adherence assessment; serial measures of TFV-DP in whole blood suggest 41% to 47% of women took sufficient periconception PrEP to prevent HIV. These data suggest that women planning for and with pregnancy should be prioritized for PrEP implementation, particularly in settings with high fertility rates and generalized HIV epidemics. Future iterations of this work should compare the outcomes to current standard of care.

## Trial registration

ClinicalTrials.gov Identifier: NCT03832530 https://clinicaltrials.gov/ct2/show/NCT03832530?term=lynn+matthews&cond=hiv&cntry=UG&draw=2&rank=1.

## Author summary

### Why was this study done?

- Tenofovir (TFV) disoproxil fumarate/emtricitabine as pre-exposure prophylaxis (PrEP) prevents HIV, is safe during pregnancy and breastfeeding, and is recommended by the WHO, CDC, and the Uganda Ministry of Health (MOH) for people exposed to HIV, including during periconception, pregnancy, and postpartum.

- Many women choose to conceive with partners who may be living with HIV but adherence to daily oral PrEP can be challenging.

- We designed and tested a 3-session Healthy Families-PrEP intervention in a rural Ugandan hospital HIV clinic and evaluated adherence to PrEP during periconception and pregnancy.

### What did the researchers do and find?

- Women planning pregnancy were counseled on ways to have a child while avoiding HIV acquisition, offered PrEP, and, for those choosing PrEP, provided with quarterly adherence support.

- Approximately 131 women enrolled, 90% initiated PrEP and, among those, 85% took at least 80% of doses (measured via electronic pillcap) over 9 months. For women continuing PrEP in pregnancy, adherence persisted, by pillcap, over 9 months.

- Plasma TFV concentrations were high among 66%, 56%, and 45% at months 3, 6, and 9, respectively. TFV-DP concentrations were high among 47%, 41%, and 45% of women at months 3, 6, and 9, respectively.

### What do these findings mean?

- Women in Uganda with PrEP indications and planning for pregnancy chose to use PrEP and most were able to sustain high adherence to daily oral PrEP prior to and during pregnancy.

- These findings suggest that women planning for and with pregnancy should be prioritized for PrEP implementation, particularly in settings with high fertility rates and generalized HIV epidemics.

## Introduction

Despite declining overall HIV prevalence in Uganda, median antenatal HIV prevalence remains high at 6% to 7% [1]. Uganda has one of the highest total fertility rates in the world at 4.7 children per woman [2], and while services to prevent perinatal transmission are robust for pregnant women with HIV, HIV prevention prior to a desired pregnancy is rarely addressed. However, at least 30% to 50% of men living with HIV in Uganda desire children [3–7] and nearly half have a stable, HIV–negative partner [8]. Women risk condomless sex to meet

important personal and sociocultural goals to have children [3,7,9–11]. In 2019, over 20,000 women of reproductive age and 5,700 children were newly diagnosed with HIV in Uganda, with perinatal transmission accounting for most infections [12]. Integrating HIV prevention into reproductive health programs presents an opportunity to reduce HIV incidence among women and infants in settings where fertility rates and HIV prevalence are high. Indeed, prevention in this context is relevant for many women across sub-Saharan Africa where the average fertility rate is 4.6 [13,14].

Effective HIV prevention strategies are available to women who want to conceive with a partner living with HIV including delaying condomless sex until the partner achieves viral load suppression by taking antiretroviral therapy (ART), treating sexually transmitted infections (STIs) in both partners, limiting condomless sex to peak fertility, and semen processing [15–17]. However, in settings where gender power imbalances make it challenging for a woman to insist that her partner participate in strategies to reduce sexual HIV transmission and where many men are not aware of their HIV-serostatus, pre-exposure prophylaxis (PrEP) is an important safer conception strategy. Data suggest that tenofovir disoproxil fumarate (TDF) and emtricitabine (FTC) are safe to use during early pregnancy, and with high adherence, PrEP can nearly eliminate HIV acquisition risks [18–20].

The World Health Organization and the Ugandan Ministry of Health (MoH) recommend PrEP as a preventive approach for HIV–negative individuals at high-risk of acquiring HIV, including women partnered with someone living with HIV [21,22]. PrEP implementation in periconception and antenatal settings has been low [23], and data on PrEP uptake and adherence among women planning for pregnancy are limited [24,25]. The few observational studies and clinical trials exploring PrEP use and adherence among specific populations of women, including adolescent girls, young women, and those in HIV-serodifferent couples, in East and Southern Africa document varying acceptability and uptake of PrEP ranging from 8% among adolescent girls to 100% among sexually active women without stated plans to become pregnant [26–30]. PrEP adherence studies of women at risk for HIV during pregnancy also observe variable adherence rates based on pharmacy pick-up as well as drug levels, ranging from 22% to 62% at 3 months [28,31,32]. Understanding how women initiate and adhere to PrEP as periconception risk reduction is an important step towards developing comprehensive HIV prevention care for women of reproductive age [33,34].

Based on formative work and informed by a conceptual framework for periconception HIV-exposure behavior [35], we developed a counseling support intervention, Healthy Families-PrEP, for HIV–negative women of reproductive age in Uganda with personal or partner plans for a pregnancy in the next year. Healthy Families-PrEP leverages individual- and couple-level reproductive goals to promote uptake and use of HIV prevention strategies, including TDF/FTC PrEP. In prior publications, we described high client- and provider-level acceptability of integrating the counseling program into routine HIV care [36]. Here, we show how women participating in this combination counseling intervention used PrEP during periconception and pregnancy periods.

## Methods

### Study design and population

Women enrolled between June 2017 and January 2019. Recruitment took place in rural, southwestern Uganda from a safer conception pilot program located within the Mbarara Regional Referral Hospital [36], HIV counseling and testing sites in the district, and via referrals from local healthcare providers. Women also approached the program after hearing about it via flyers, community testing events, and informational radio spots.

Eligible women were aged 18 to 40 years, tested negative for HIV (rapid test), not currently pregnant (urine β-HCG testing), likely to be fertile (based on reproductive health history) [37], and reported personal or partner desire to have a child in the next year [38–41]. Additionally, an eligible woman either knew her pregnancy partner was living with HIV or felt she was at risk for acquiring HIV based on the Perceived Risk of HIV Scale [42]. All enrolled women provided written informed consent and felt able to attend study visits for the duration of the study.

### Study procedures

The protocol and analysis procedures were defined prior to participant enrollment [43]. Participants received a package of HIV prevention (or safer conception) counseling for women who want to conceive a child while exposed to HIV, Healthy-Families PrEP (Fig 1). Study visits occurred quarterly and included HIV testing, pregnancy testing, PrEP adherence counseling, and safer conception counseling sessions.

The Healthy Families-PrEP intervention was informed by a socio-ecological framework to understand women's PrEP use [44,45] and our periconception risk behavior conceptual framework [35]. Healthy Families-PrEP leveraged individual- and couple-level reproductive goals to promote uptake and use of HIV prevention strategies through education, problem-solving, communication skills training, and adherence support. Counselors completed a 2-day in-person training (together with a research team in South Africa [46]) and quarterly supervision calls with members of the study team (CP and postdoctoral fellows). Trained counselors

---

## Healthy Families-PrEP Intervention Content

### Elements of Healthy Families-PrEP Plan

Initiate PrEP, Adhere to PrEP (quarterly adherence support)
Couples HIV Counseling & Testing
Condoms / contraception until partner serostatus known / VL suppressed

| Session 1 | Sessions 2-3 |
|---|---|
| (60 minutes) | (30 min/each)(every 3 months) |
| Rapport building & Introduction<br>Safer conception education<br>Motivational strategies to prepare for behavior change<br>Develop individualized safer conception plan* | Review prior session content<br>Safer conception plan review<br>Revise plan as needed<br>Support to implement plan |

### Methods employed to help participant implement plan
Education  +  Problem solving + Motivational strategies + Communication skills

←———— **12 weeks** from enrollment to completion of all sessions ————→

*Women who chose to use PrEP received quarterly adherence support based on the Lifesteps (Psaros et al., 2014) intervention through the end of study follow up.*

**Fig 1. Intervention content from the Healthy Families-PrEP Program.**

used an intervention guide and were directed to complete a 60-min introductory session followed by 30-min follow-up sessions in which they worked with participants to develop and implement a safer conception plan. For women who did not know their partner's HIV status, this plan included how to encourage their partners to test and disclose their status. For those with partners living with HIV, counseling included how to encourage partners to initiate ART, to delay condomless sex until partner achieved either ART coverage for 6 months or HIV viral load suppression, and to limit sex without condoms to peak fertility. Oral combination TDF (300 mg)/FTC (200 mg) as Truvada was offered as PrEP and additional adherence counseling support was included at quarterly visits as long as the participant chose to use PrEP based on methods developed by Dr. Psaros and colleagues and included in the CDC compendium of evidence-based interventions [47,48].

Support for each of these strategies was available at the clinic study site including couples-based counseling and testing, condoms, ART, and contraception. Ovulation prediction kits were offered by the study. In addition, participants were counseled regarding opportunities for sperm washing, donor sperm, and adoption as alternatives available in other parts of Uganda.

At baseline, women completed a questionnaire that included measures on sociodemographic, health status, reproductive history, HIV knowledge, safer conception behaviors, and other constructs expected to impact PrEP uptake and adherence based on our periconception risk reduction conceptual framework [35]. Women were eligible to initiate TDF/FTC as PrEP at any time during study follow-up. Through April 2019, women who became pregnant during follow-up had final study evaluations at the time of first positive pregnancy test. They were exited from the study with referrals for antenatal and routine PrEP care. Based on ethical concerns about pregnant women not wanting to access PrEP through the public sector, the protocol was updated. From March 2019, women who became pregnant during study follow-up remained in the study (where PrEP was provided to women choosing to use it) and were followed every 3 months until a pregnancy outcome occurred. For women with incident pregnancies after March 2019, the final visit was conducted after the pregnancy outcome. At all final visits, women were referred for PrEP care in the public sector if desired. Women who tested positive for HIV during study follow up completed exit activities at the time of first positive HIV test and were referred to appropriate HIV follow-up care. Women were categorized as lost-to-follow-up after missing 2 or more consecutive scheduled study visits and/or failure to attend further study visits after at least 2 documented attempts to contact or locate the participant.

All participants completed quarterly urine pregnancy tests (beta-HCG), rapid HIV1/2 antibody screening (and confirmation as indicated) per Ugandan standard of care, and syndromic screening for STIs. A subset of women was screened for asymptomatic STI including *Chlamydia trachomatis*, *Neisseria gonorrhoeae*, and *Trichomonas vaginalis* via GeneXpert testing and syphilis via treponemal and non-treponemal antibody testing at baseline and 6 months follow-up. Participants who screened or tested positive for STIs received treatment per local guidelines; STI prevalence and incidence findings are reported elsewhere [49]. Blood was drawn at baseline for creatinine and hepatitis B assessment to ensure no contraindications to PrEP use. Creatinine concentrations were assessed during quarterly follow-up. Women with abnormal renal function (serum creatinine >89 μmol/L and/or GFR <60 mL/min estimated using the Cockcroft–Gault equation) or active hepatitis B infection (HBV surface antigen positive) were subsequently instructed to discontinue PrEP.

## Measures

As part of questionnaires, data were collected on sociodemographic characteristics (age, education, and socioeconomic status), reproductive health history (number of prior pregnancies,

live births and living children), sexual behavior (number of sexual partners, condom use, and sexual encounters), and HIV disclosure within the pregnancy partnership. Questionnaire responses were collected to construct the following scores: knowledge of HIV [50] and of safer conception behaviors [51], HIV risk perception [42], PrEP optimism (adapted) [52], parenthood motivation [53], reproductive autonomy [54], sexual relationship power [55], partner communication [56], functional social support [57], and depression [58,59]. Summary scores were derived using cited methods (S1 Text).

## Assessment of PrEP uptake and adherence

To measure daily pill-taking behavior, women were provided with an electronic pillbox (Wisepill Technologies, South Africa) that stored PrEP tablets and recorded when the device was opened, providing a reliable, objective assessment of day-to-day adherence behavior [60]. We assessed 2 primary outcomes: (1) uptake of PrEP defined as the proportion of enrolled women who ever initiated PrEP; and (2) objective adherence to PrEP as measured by the proportion of days with pillbox opening during the first 3 months of active PrEP follow-up among PrEP initiators. Adherence was defined as the number of days with a time-stamped record of a device opening divided by the number of days the participant was in active PrEP follow-up (defined as PrEP initiation through to the earlier of reported PrEP discontinuation or study exit) and capped at 1 opening per day. We chose to focus on 3-month adherence given high pregnancy incidence resulting in meaningful changes in the cohort over time. We also report on mean monthly adherence over time and the proportion of PrEP initiators with monthly adherence ≥80%, categorized as "high" adherence [61,62]. These data are also described for women with pregnancy as secondary analyses. (We did not analyze frequency of or factors associated with 72 h gaps as per the analysis plan given low frequency of 72 h gaps in our data and evolution in the understandings of PrEP to recognize that these gaps may not be as meaningful for PrEP use adherence.)

We used multiple approaches to estimate PrEP adherence given that all adherence measures have advantages and limitations. Specifically, electronic monitoring provides highly informative day-to-day patterns of adherence; however, it may under or overestimate adherence due to device non-use or curiosity openings of the pillbox without dosing, respectively. Biomedical assessments of PrEP adherence provide documentation of pill ingestion, yet they can only be collected at periodic time points due to resource and logistical constraints, and interpretation of drug concentrations may be complicated by variable metabolism, drug interactions, and/or laboratory variation. Notably, the prior time period reflected by each measure varies (i.e., approximately 7 days for plasma, approximately 8 weeks for erythrocytes after achievement of steady state). We therefore determined plasma TFV and dried blood spot (DBS) intraerythrocytic TFV-DP concentrations for a subset of participants.

Plasma TFV concentrations ≥40 ng/mL indicate dosing in the last 24 h (categorized as "high" adherence), between 10 and 40 ng/mL indicates dosing in the last 3 days (categorized as "moderate" adherence), and between >0.31 and 10 ng/mL indicates dosing in the last week (categorized as "low" adherence) [20,63]. TFV concentrations below 0.31 ng/mL are below the limits of detection. In contrast to data from men [64–67], data on how to assign categories of TFV-DP concentrations to pill-taking behavior for women who are not pregnant nor postpartum women in African settings is evolving [66,68]. Based on the available data and the expertise on this authorship team, ≥600 fmol/punch was categorized as indicating approximately ≥4 doses per week (categorized as high adherence), 450 to 600 fmol/punch indicates about 3 doses per week (categorized as moderate adherence), and between 31.3 and <450 indicates <3 doses per week (categorized as low adherence) [20,60,67,69]. TFV-DP concentrations below <31.3 fmol/punch are below limits of detection.

## Power and sample size

We proposed to enroll 150 women to have an evaluable sample of 112 women initiating PrEP. Our estimated evaluable sample size was based on preliminary data suggesting that 75% of women will choose to initiate PrEP. However, challenges with recruitment related to national stock-outs of HIV testing supplies resulted in halting enrollment prior to reaching this sample size. However, due to higher than anticipated PrEP use, we had sufficient sample size to evaluate factors associated with PrEP adherence using the continuous outcome measure of electronic pillcap adherence.

## Statistical analysis

We conducted univariable and multivariable-adjusted analyses to assess predictors of mean PrEP adherence during the first 3 months following PrEP initiation. Given our smaller than expected sample size, we chose to model factors associated with adherence by electronic pillcap (rather than plasma TFV $> = 40$ ng/mL as per original analysis plan) given more complete data, a continuous outcome, and good correlation between plasma TFV and electronic pillcap data (see Results). Baseline covariates were selected based on our periconception HIV risk conceptual framework [35] and included age, education, number of live births, depression, parenthood motivation (as related to social control), sexual relationship power scale, reproductive autonomy (decision-making subscale), and perceived HIV risk. For each baseline predictor (relative to the adherence outcome), we separately constructed multivariable-adjusted linear regression models with generalized estimating equations (GEEs) using a change-in-estimate approach [70]. Specifically, relative to a fully adjusted model (i.e., all known and/or hypothesized confounding factors identified by subject matter knowledge of the underlying causal structure included), we removed, one by one, each factor and recorded the estimated change in percentage adherence for the predictor–outcome association of interest. If the removal of the factor changed the change in percentage adherence by $\geq 10\%$, it was retained in the final multivariable model. Given the large number of results only the final adjusted change in percentage adherence (and 95% CI) for each predictor of interest is presented. A complete description of each multivariable-adjusted model is included in S2 Text. We note that our original analysis plan specified log-binomial regression with GEE for estimation of relative risks, respectively. We used linear regression with GEE as is appropriate for a continuous outcome.

For secondary analyses of adherence over 6 and 9 months of follow-up, we fit intercept only linear regression models with GEE stratified by month to estimate mean monthly adherence and 95% CI. Overall, adherence over 6 and 9 months was calculated by taking an inverse variance weighted mean of monthly estimates to account for variable amounts of person-time contributing to each monthly estimate. We performed Spearman correlation analysis to assess the relationship between plasma TFV and whole blood drug TFV-DP concentrations and electronic pillcap adherence at each of the 3-month follow-up visits (through 9 months) where data was available. Lastly, we performed exploratory analyses to assess adherence to PrEP before and after date of first positive pregnancy test until the reported date of pregnancy outcome. All statistical analyses were conducted using SAS software version 9.4 (SAS Institute, Cary, North Carolina, United States of America).

## Ethics statement

Ethics approvals were secured from the Partners HealthCare Institutional Review Board (IRB), Mbarara University of Science and Technology Research Ethics Committee, and University of Alabama at Birmingham IRB. Regulatory approvals were also secured from Uganda's Office of the President and the National Council of Science and Technology.

## Results

Of the 916 women who were screened, 131 (14%) met study criteria and enrolled. The study was designed to enroll 150 women, but accrual took longer than expected in part due to recurrent stock-outs of HIV testing supplies in Uganda. Reasons for ineligibility are shown in Fig 2.

Table 1 presents the baseline characteristics of enrolled participants. Median (25th to 75th percentile) age was 28.7 (27.8 to 29.5) years with nearly all women (92%) reporting a prior pregnancy. Among women with a prior pregnancy, median (25th to 75th percentile) number of pregnancies was 2.9 (2.7 to 3.2). Most participants had completed primary school (68%), were currently employed (73%), married or living as married (94%), and reported that their partner or spouse was living with HIV ($N$ = 97; 74%). Seventy-nine (60%) reported

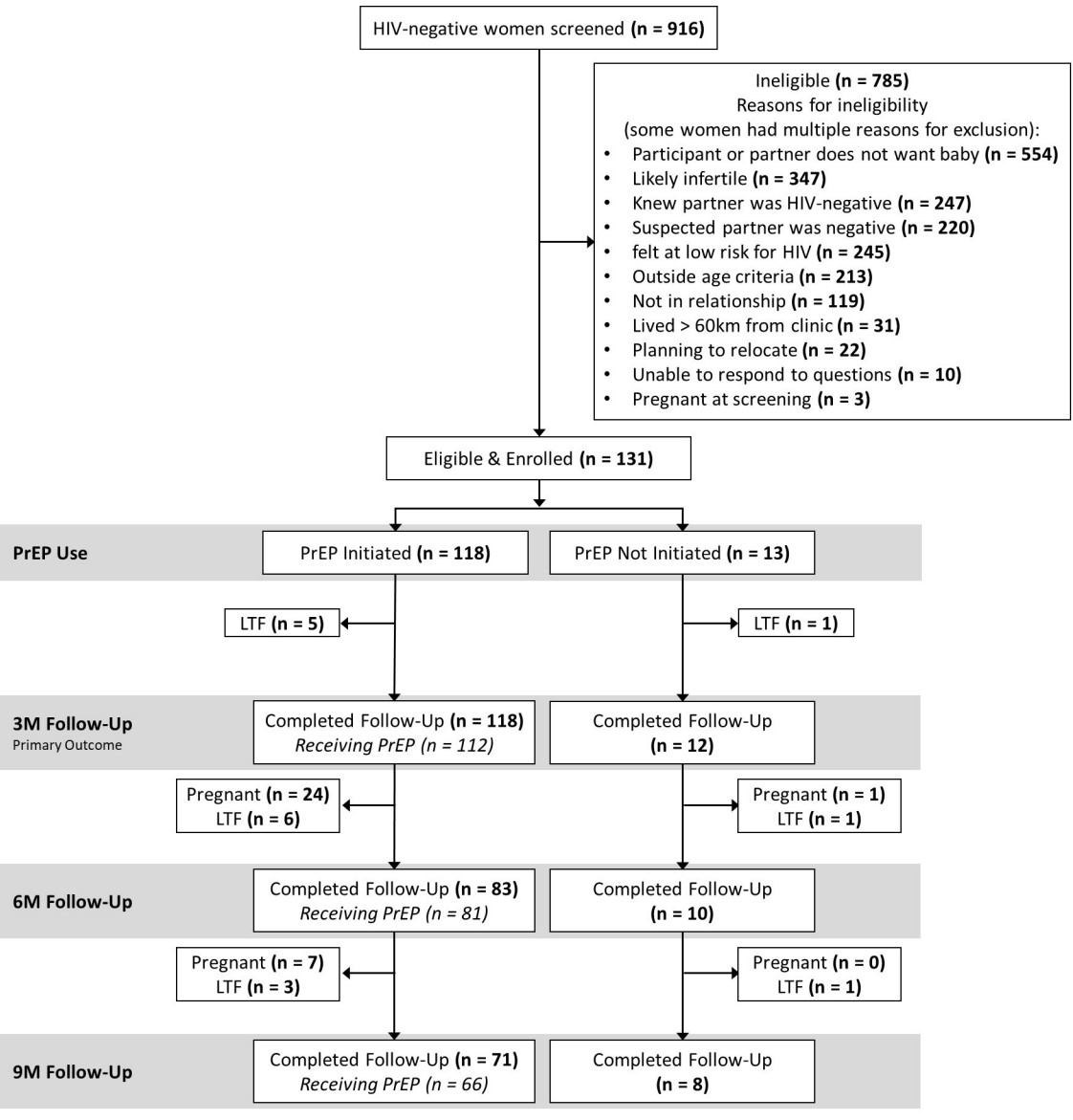

LTF: Lost to Follow-Up

**Fig 2. Summary of screening, eligibility, enrollment, PrEP uptake, and follow-up.**

**Table 1.  Baseline characteristics of *N* = 131 participants in the Uganda Healthy Family Study, overall and according to PrEP uptake.**

| Participant characteristics | | PrEP uptake | |
|---|---|---|---|
| | Overall (*N* = 131) N (%) | Initiated PrEP (*N* = 118) N (%) | Did not initiate PrEP (*N* = 13) N (%) |
| **Age, years (*n* = 131)** | | | |
| 17.4 < 25.9 | 43 (33%) | 37 (31%) | 6 (46%) |
| 25.9 < 30.9 | 43 (33%) | 38 (32%) | 5 (38%) |
| 30.9–39.8 | 45 (34%) | 43 (36%) | 2 (15%) |
| **Education (*n* = 131)** | | | |
| None to some primary | 40 (31%) | 37 (31%) | 3 (23%) |
| Completed primary | 24 (18%) | 23 (19%) | 1 (8%) |
| Early secondary (S1–S3) | 24 (18%) | 21 (18%) | 3 (23%) |
| Later to completed secondary (S4 or S6) | 32 (24%) | 27 (23%) | 5 (38%) |
| Tertiary/vocation, university | 11 (8%) | 10 (8%) | 1 (8%) |
| **Currently employed (*n* = 131)** | 96 (73%) | 86 (73%) | 10 (77%) |
| **Income, past 3 months (*n* = 131)** | | | |
| 0–140,000 UGX | 65 (50%) | 59 (50%) | 6 (46%) |
| 150,000–290,000 UGX | 26 (20%) | 24 (20%) | 2 (15%) |
| 300,000–490,000 UGX | 20 (15%) | 16 (14%) | 4 (31%) |
| 500,000–3,000,000 UGX | 20 (15%) | 19 (16%) | 1 (8%) |
| **Participant (or member of household) owns. . .** | | | |
| . . . a house (*n* = 131) | 58 (44%) | 53 (45%) | 5 (38%) |
| . . . any land (*n* = 131) | 84 (64%) | 76 (64%) | 8 (62%) |
| . . . any livestock (*n* = 131) | 50 (38%) | 44 (37%) | 6 (46%) |
| **Reproductive history** | | | |
| **Ever pregnant (*n* = 131)** | 121 (92%) | 110 (93%) | 11 (85%) |
| **Number of pregnancies (*n* = 131)** | | | |
| 0 | 10 (8%) | 8 (7%) | 2 (15%) |
| 1 | 21 (16%) | 20 (17%) | 1 (8%) |
| 2 | 34 (26%) | 27 (23%) | 7 (54%) |
| 3 | 30 (23%) | 29 (25%) | 1 (8%) |
| 4 or more | 36 (27%) | 34 (29%) | 2 (15%) |
| **Number of live births[1] (*n* = 121)** | | | |
| 0 | 7 (6%) | 7 (6%) | 0 (0%) |
| 1 | 32 (26%) | 29 (26%) | 3 (27%) |
| 2 | 44 (36%) | 38 (35%) | 6 (55%) |
| 3 | 21 (17%) | 20 (18%) | 1 (9%) |
| 4 or more | 17 (14%) | 16 (15%) | 1 (9%) |
| **Problematic drinking past year[2] (*n* = 130)** | 14 (11%) | 13 (11%) | 1 (8%) |
| **Depression (>1.75) (*n* = 131)** | 46 (35%) | 42 (36%) | 4 (31%) |
| **Number of sexual partners, past 3 months (*n* = 131)** | | | |
| 0 | 4 (3%) | 4 (3%) | 0 (0%) |
| 1 | 115 (88%) | 102 (86%) | 13 (100%) |
| 2 or more | 12 (9%) | 12 (10%) | 0 (0%) |
| **Marital status (*n* = 127)** | | | |
| Spouse/legal partner | 98 (77%) | 86 (75%) | 12 (92%) |
| Living as married | 22 (17%) | 22 (19%) | 0 (0%) |
| Long-term partner | 7 (6%) | 6 (5%) | 1 (8%) |
| **Condom use during last sex with main partner (*n* = 127)** | 48 (38%) | 43 (38%) | 5 (38%) |

*(Continued)*

**Table 1.** (Continued)

| Participant characteristics | | PrEP uptake | |
|---|---|---|---|
| | Overall (*N* = 131) *N* (%) | Initiated PrEP (*N* = 118) *N* (%) | Did not initiate PrEP (*N* = 13) *N* (%) |
| **Any condom use with main partner (*n* = 127)** | | | |
| No | 97 (76%) | 87 (76%) | 10 (77%) |
| Yes | 30 (24%) | 27 (24%) | 3 (23%) |
| **HIV status of main pregnancy partner (*n* = 131)** | | | |
| No partner | 4 (3%) | 4 (3%) | 0 (0%) |
| Unknown partner | 29 (22%) | 23 (19%) | 6 (46%) |
| HIV–negative | 1 (1%) | 0 (0%) | 1 (8%) |
| HIV–positive | 97 (74%) | 91 (77%) | 6 (46%) |
| **Parenthood motivation, mean (95% CI)[3] (*n* = 131)** | | | |
| Parenthood motivation subscale: Happiness | 8.6 (8.5, 8.7) | 8.6 (8.5, 8.7) | 8.6 (8.1, 9.1) |
| Parenthood motivation subscale: Well-being | 8.4 (8.3, 8.6) | 8.4 (8.3, 8.6) | 8.5 (8.0, 8.9) |
| Parenthood motivation subscale: Identity | 8.1 (7.9, 8.4) | 8.1 (7.9, 8.4) | 8.3 (7.8, 8.8) |
| Parenthood motivation subscale: Parenthood | 7.9 (7.6, 8.1) | 7.9 (7.6, 8.1) | 7.9 (7.3, 8.5) |
| Parenthood motivation subscale: Social control | 7.1 (6.8, 7.5) | 7.2 (6.8, 7.5) | 7.0 (5.9, 8.1) |
| Parenthood motivation subscale: Continuity | 8.7 (8.6, 8.8) | 8.7 (8.6, 8.8) | 8.9 (8.8, 9.1) |
| **PrEP optimism, mean (95% CI)[4] (*n* = 19)** | 6.5 (5.7, 7.2) | 6.6 (5.9, 7.4) | 5.0 (1.1, 8.9) |
| **Sexual relationship power scale, mean (95% CI)[5] (*n* = 131)** | 2.3 (2.2, 2.4) | 2.3 (2.2, 2.4) | 1.9 (1.6, 2.3) |
| **Reproductive autonomy, mean (95% CI)[6] (*n* = 131)** | | | |
| Reproductive autonomy subscale: Free from coercion | 2.7 (2.6, 2.7) | 2.7 (2.6, 2.8) | 2.4 (2.1, 2.7) |
| Reproductive autonomy subscale: Communication | 2.8 (2.7, 2.8) | 2.8 (2.7, 2.8) | 2.8 (2.6, 3.0) |
| Reproductive autonomy subscale: Decision-making | 2.0 (1.9, 2.1) | 2.0 (1.9, 2.1) | 1.9 (1.6, 2.1) |
| **Perceived HIV risk, mean (95% CI)[7] (*n* = 131)** | 21.3 (20.9, 21.8) | 21.3 (20.8, 21.8) | 21.9 (20.3, 23.5) |
| **HIV knowledge, mean (95% CI)[8] (*n* = 131)** | 15.2 (14.6, 15.8) | 15.3 (14.7, 15.9) | 14.2 (12.3, 16.0) |
| **Social support, mean (95% CI)[9] (*n* = 131)** | 3.2 (3.1, 3.3) | 3.2 (3.1, 3.3) | 3.3 (3.0, 3.6) |

[1]Number of live births reported among those who were pregnant at least once.

[2]Problematic drinking is defined as any who indicate at least one of the following: feel guilty after drinking, no memory of actions while drinking, failed to do what was expected due to drinking, and/or drink when first getting up.

[3]Responses related to parenthood motivation were assigned values of 1 (for "disagree"), 2 (for "partially agree"), or 3 (for "strongly agree") (S1 Text).

[4]Participants responded to 3 statements of PrEP optimism: (1) "PrEP reduces risk of getting HIV;" (2) "PrEP makes it easier to relax about sex without condoms;" and (3) "PrEP makes me worry less." Responses to these statements were assigned values of 0 (for "strongly disagree"), 1 (for "disagree"), 2 (for "agree"), 3 (for "strongly agree"), and summed to create a score. Higher scores indicate greater PrEP optimism.

[5]Higher score indicates more power in relationship including average of relationship control and decision-making dominance scores.

[6]Higher score indicates greater reproductive autonomy for participants, summing over the responses for 3 subscales (S1 Text).

[7]Sum of 6 individual questions related to perceived risk of acquiring HIV (higher score -> higher felt risk) (S1 Text).

[8]Based on series of "True/False" questions testing knowledge of HIV. Correct answers given score of 1; score based on sum of correct answers (out of 24 questions). Examples of these questions are "Coughing and sneezing DO NOT spread HIV" and "If a woman has HIV then her baby will always be born with HIV."

[9]There 10 statements related to social support with responses scored as (1) "Never," (2) "Much less than I would like," (3) "Less than I would like," and (4) "As much as I would like" were averaged to create a social support score. Higher scores correspond to higher felt support. Examples of these statements are "I get visits from friends and relatives" and "I get help with money in an emergency."

condomless sex at last encounter. Forty-five (35%) women screened positive for symptoms of depression, 14 (11%) reported problematic drinking within the past year, and perceived HIV risk score was high (21.3, 95% CI: 20.9 to 21.8).

Among 131 enrolled women, 17 (13%) moved or were otherwise lost to follow-up. A total of 53 pregnancies occurred among 131 women with 848 person-months of follow-up (median

8.8 months), resulting in an incidence of 75 pregnancies per 100 person-years (95% CI: 57, 98) for a 1-year cumulative incidence of 53% (95% CI: 43%, 62%) (Fig 2).

## PrEP uptake

Of the 131 women enrolled, a total of 118 women (90%) initiated PrEP, all of whom chose to initiate at baseline. None had hepatitis B infection nor were ineligible based on serum creatinine. Those who initiated PrEP were older (mean age 29.0 versus 25.7 years) and more likely to report having ≥3 lifetime pregnancies compared to those who did not initiate PrEP (54% versus 23%). A greater proportion of women who initiated PrEP reported their partner was living with HIV compared to the proportion of women who did not initiate PrEP (76% versus 46%). Mean perceived HIV risk score was lower among women who initiated PrEP compared to those who did not initiate PrEP (21.3 versus 21.9) and women who initiated PrEP in this study had higher PrEP optimistic beliefs compared to those that did not initiate PrEP (Table 1).

## Periconception PrEP adherence

Among PrEP initiators, 101 (86% of $N = 118$) had electronic adherence data through the first 3 months (Fig 2). Among these women, average adherence was 87% (95% CI: 83%, 90%) and 86 (85%) women had high adherence (≥80% of expected doses taken) through 3 months. PrEP adherence was not significantly associated with covariates of interest in adjusted models (Table 2, full model details in S2 Text). Monthly adherence through 9 months was consistently high (Fig 3): in the first month, average adherence was 86% ($N = 107$; 95% CI: 83%, 90%) compared to the final month average adherence of 90% ($N = 48$; 95% CI: 85%, 95%). Longitudinal models showed that month on study was not associated with adherence (S3 Text). No women had provider-directed PrEP holds or stops for renal dysfunction or other clinical events.

## Periconception drug concentrations

TFV concentrations were processed from 112 participants contributing 44 plasma and 104 DBS samples at 3 months. At 6 months, 25 plasma and 79 DBS samples collected from 81 participants were processed. At 9 months, 22 plasma and 65 DBS samples collected from 66 participants were processed. All processed samples were from women who were not pregnant at the time of collection.

Plasma TFV ≥40 ng/mL was observed in 66%, 56%, and 45% of samples collected at 3, 6, and 9 months, respectively (Fig 4). Electronic adherence data from 3 and 30 days prior to sample collection were significantly correlated with plasma TFV levels at the 3-month visit ($\rho$ = 0.45; $P$ = 0.006 for 3 days and $\rho$ = 0.44; $P$ = 0.01 for 30 days). Correlations at 6 and 9 months were also moderate to high and statistically significant (S3 Text), suggesting that pill-taking behavior correlates well with objective adherence behavior over the past week. TFV-DP (indicative of use during past 6 to 8 weeks) concentrations of ≥600 fmol/punch were detected among 47% of 3-month, 41% of 6-month, and 45% of 9-month samples (Fig 4). Electronic adherence data and 3-month TFV-DP levels did not correlate with pillcap data collected during the 30 ($\rho$ = 0.06; $P$ = 0.55) and 60 ($\rho$ = 0.17; $P$ = 0.11) days prior to sample collection. Correlations at 6 and 9 months were similarly low and not statistically significant (S3 Text). Fig 4 presents the proportion of participants with high, medium, low, and undetected adherence by pillcap, plasma, and DBS at 3, 6, and 9 months of follow-up.

**Table 2. Univariable and multivariable-adjusted change in mean (95% CI) adherence[a] to PrEP during 3 months following initiation.**

| | Univariable | | Multivariable-adjusted[b] | |
|---|---|---|---|---|
| Covariate | change in percent adherence (95% CI) | P | Change in percent adherence (95% CI) | P |
| Age per 5 years | 1.7% (−1.7%, 5.1%) | 0.33 | 1.7% (−1.7%, 5.1%) | 0.33 |
| Some secondary education or higher (vs. none or primary education) | 2.4% (−4.6%, 9.4%) | 0.50 | 2.1% (−4.8%, 9.0%) | 0.55 |
| Number of live births (2+ vs. 0,1) | 2.5% (−4.8%, 9.7%) | 0.50 | 1.0% (−6.9%, 8.9%) | 0.80 |
| Depression score (>1.75 vs. < = 1.75) | −5.6% (−12.7%, 1.5%) | 0.12 | −5.6% (−12.7%, 1.5%) | 0.12 |
| Parenthood motivation subscale: Social control | 0.04% (−1.8%, 1.9%) | 0.97 | 0.04% (−1.9%, 2.0%) | 0.97 |
| Sexual relationship power scale | 1.1% (−4.2%, 6.5%) | 0.68 | 0.4% (−5.1%, 5.9%) | 0.89 |
| Reproductive autonomy subscale: Decision-making | −2.6% (−9.5%, 4.3%) | 0.46 | −3.2% (−10.2%, 3.7%) | 0.36 |
| Perceived HIV risk score | −0.1% (−1.3%, 1.2%) | 0.90 | 0.04% (−1.2%, 1.3%) | 0.95 |

[a]Adherence measured as the percentage of time-stamped pillbox openings during active PrEP follow-up.

[b]Multivariable-adjusted associations estimated using a covariate adjusted for confounders determined to be of interest using change-in-estimate methods considering all other covariates; final models for each covariate are included in the Supporting information.

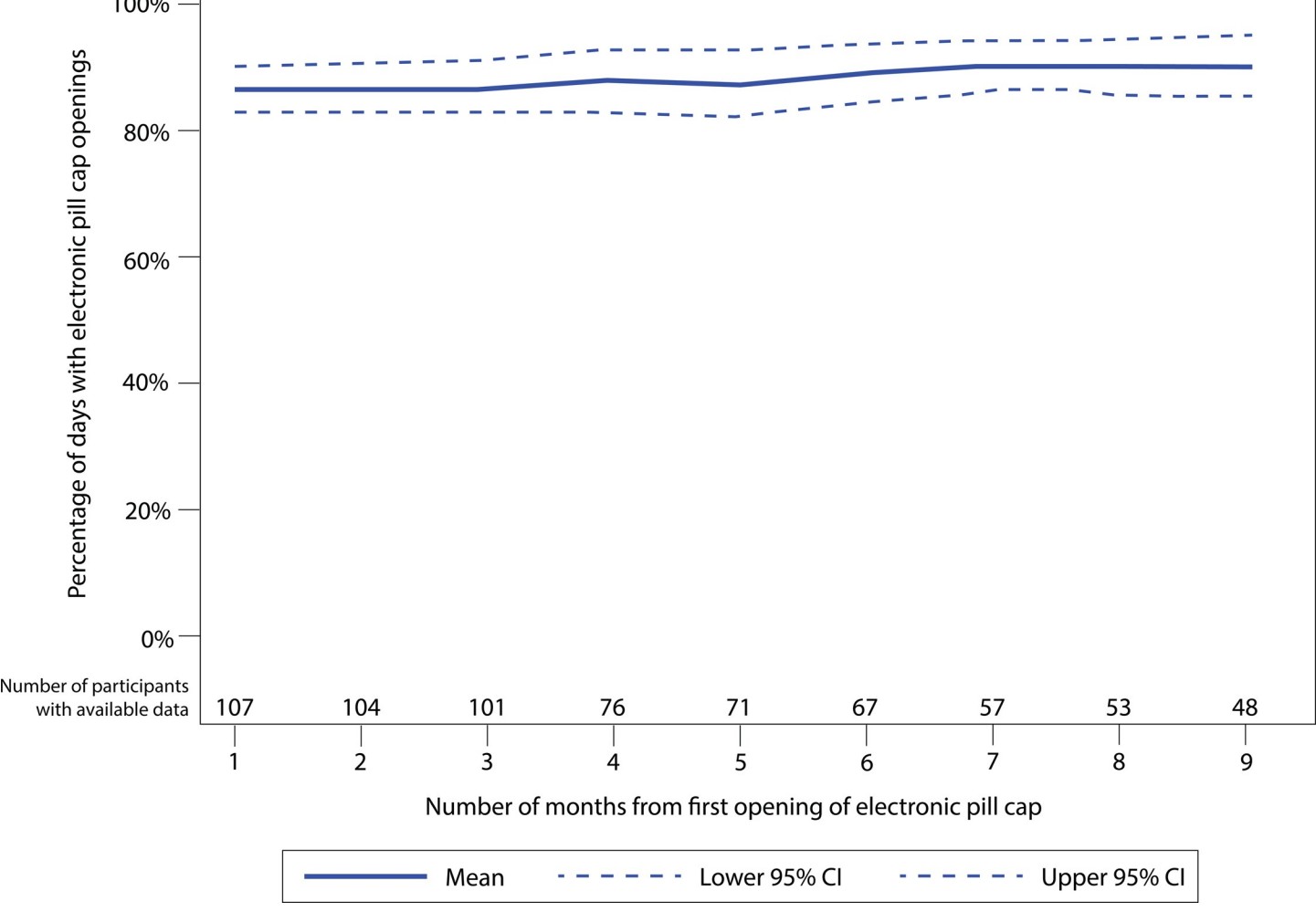

**Fig 3. Mean and 95% confidence interval bands for electronic adherence to PrEP during periconception period over time.**

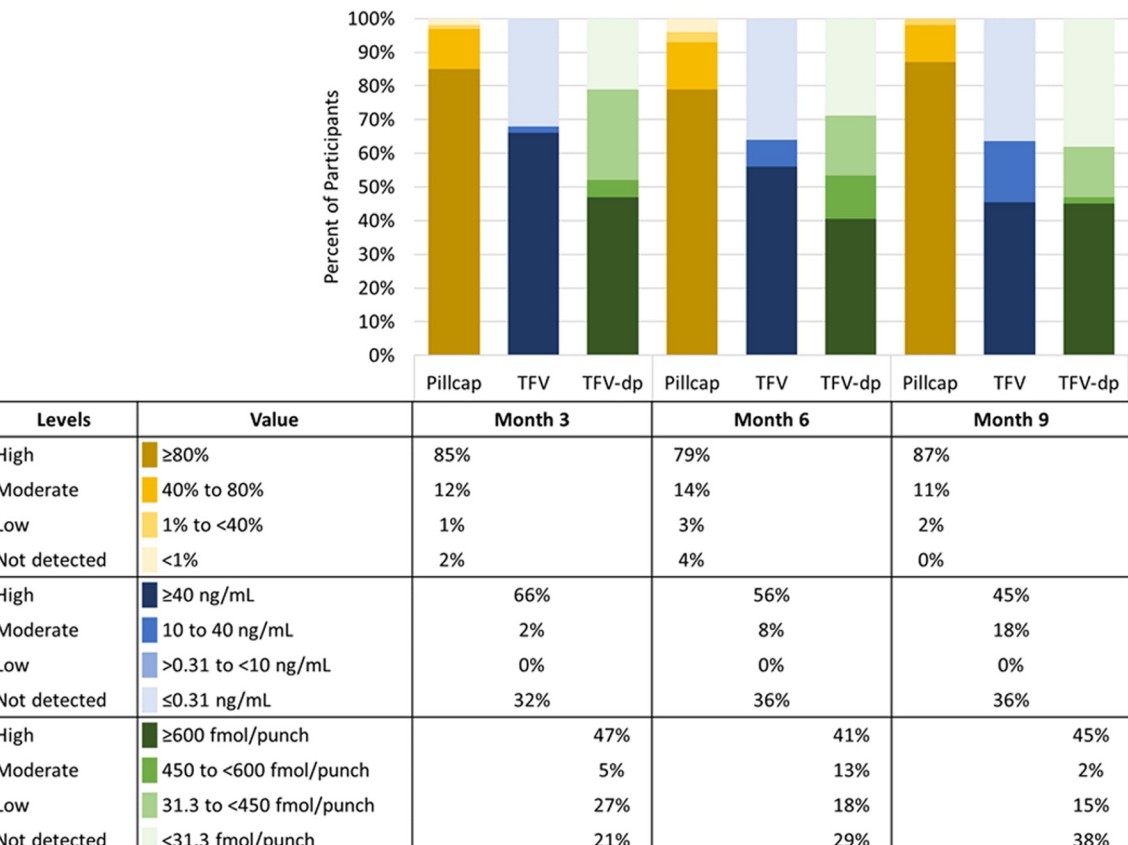

| Levels | Value | Month 3 | | | Month 6 | | | Month 9 | | |
|---|---|---|---|---|---|---|---|---|---|---|
| | | Pillcap | TFV | TFV-dp | Pillcap | TFV | TFV-dp | Pillcap | TFV | TFV-dp |
| High | ≥80% | 85% | | | 79% | | | 87% | | |
| Moderate | 40% to 80% | 12% | | | 14% | | | 11% | | |
| Low | 1% to <40% | 1% | | | 3% | | | 2% | | |
| Not detected | <1% | 2% | | | 4% | | | 0% | | |
| High | ≥40 ng/mL | | 66% | | | 56% | | | 45% | |
| Moderate | 10 to 40 ng/mL | | 2% | | | 8% | | | 18% | |
| Low | >0.31 to <10 ng/mL | | 0% | | | 0% | | | 0% | |
| Not detected | ≤0.31 ng/mL | | 32% | | | 36% | | | 36% | |
| High | ≥600 fmol/punch | | | 47% | | | 41% | | | 45% |
| Moderate | 450 to <600 fmol/punch | | | 5% | | | 13% | | | 2% |
| Low | 31.3 to <450 fmol/punch | | | 27% | | | 18% | | | 15% |
| Not detected | <31.3 fmol/punch | | | 21% | | | 29% | | | 38% |

**Fig 4. Periconception adherence at 3, 6, and 9 months using 3 assessment methods.**

### HIV incidence

One participant tested positive for HIV at 9 months. This participant was not pregnant. Although electronic adherence data suggested high PrEP adherence (>80% of doses all months), plasma TFV levels were undetectable at 3, 6, and 9 months of follow-up.

### Pregnancy PrEP use

Forty-eight (91%) of 53 pregnant women ever used PrEP, of whom 35 (66%) completed exit procedures at the incident pregnancy visit, 17 (32%) were followed through pregnancy out-come. Overall mean adherence by electronic pillcap during the first 6 months of pregnancy was 98% (95% CI: 97%, 99%). Fig 5 highlights consistent adherence across periconception and pregnancy time periods. Of the 17 pregnancies with outcome information, 12 (71%) resulted in live births and 5 (29%) resulted in miscarriages or stillbirth or termination.

### Discussion

We observed high PrEP uptake and use among periconception and pregnant women in Uganda participating in the Healthy Families-PrEP counseling intervention. These are the first data we are aware of showing high uptake and high, sustained adherence (by pillcap and plasma TFV) to daily, oral TDF/FTC as PrEP in a population of women with indications for HIV prevention in Uganda. Over 90% of participants chose PrEP as part of a safer conception

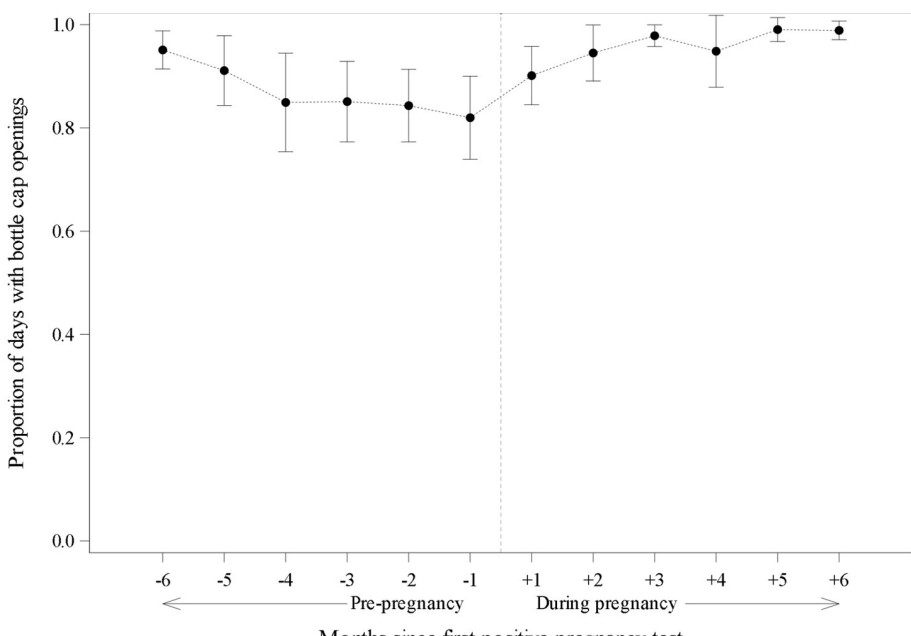

**Fig 5. Adherence prior to and post-pregnancy among those women with incident pregnancy.**

| | Months Pre-Pregnancy | | | | | | Months During Pregnancy | | | | | |
|---|---|---|---|---|---|---|---|---|---|---|---|---|
| | -6 | -5 | -4 | -3 | -2 | -1 | +1 | +2 | +3 | +4 | +5 | +6 |
| Mean | 0.95 | 0.91 | 0.85 | 0.86 | 0.84 | 0.82 | 0.90 | 0.95 | 0.98 | 0.96 | 0.99 | 0.99 |
| 95% CI | (0.92, 0.98) | (0.85, 0.97) | (0.76, 0.94) | (0.79, 0.93) | (0.78, 0.91) | (0.75, 0.90) | (0.86, 0.95) | (0.90, 0.99) | (0.96, 1.00) | (0.93, 1.00) | (0.97, 1.01) | (0.98, 1.00) |
| n | 15 | 15 | 19 | 37 | 39 | 41 | 11 | 10 | 9 | 8 | 7 | 6 |

strategy and mean pillcap adherence over 3 months was 87% (95% CI: 83%, 90%). In addition, electronic adherence demonstrated persistent adherence behavior over 9 months of periconception as well as pregnancy follow-up. Despite high pregnancy (35%) and STI (21%) incidence at 6 months [49], only a single HIV infection occurred (out of >70 person-years of follow-up). Our short-term (plasma TFV) and longer-term (intra-erythrocytic TFV-DP) data suggest that closer to half of women consistently took oral PrEP. This proportion is high compared to many other implementation studies of PrEP for women, as outlined in the introduction, and higher than observed in the TDF/FTC arm of the recent HPTN-084 trial in which 42% of women had high plasma TFV and 18% had high TFV-DP levels above 700 fmol/punch [71]; these findings highlight opportunities of PrEP for periconception and pregnancy HIV prevention. Differences observed across 3 objective measures of PrEP adherence in this cohort also emphasize the value of incorporating multiple adherence measurements that capture different time exposures while revealing gaps in our understanding of tenofovir drug-level data interpretation for cisgender African women who are neither pregnant nor postpartum.

The Healthy Families-PrEP combination intervention offered counseling delivered by local lay counselors to support HIV prevention strategies including daily PrEP use in the context of planning for or being pregnant (Fig 1). The 3 counseling sessions offered to all women were developed for individuals or couples, and we published qualitative data indicating high levels of acceptability and feasibility from the standpoint of clients and providers [36]. Quarterly adherence support was adapted from the Lifesteps intervention, an evidence-based intervention included in the CDC compendium of effective interventions for HIV prevention, and utilized education, problem solving, and motivational interviewing strategies to promote adherence. Lifesteps was first successfully adapted for PrEP among HIV-serodifferent couples

in Uganda [47]. Here, the same tenets were applied with an added emphasis on the opportunity to prevent HIV acquisition while meeting reproductive goals. We did not collect detailed process data, but counselors were trained to offer sessions lasting 30 to 60 min. Our team provided in-person training and every other month supervision support, where counselors discussed challenges. While this counseling approach was effective for most women in our cohort, some encountered ongoing adherence challenges. Future iterations of this work will aim to identify which women are most likely to experience adherence challenges, and identify what role adherence counseling may play in supporting prevention effective PrEP use, regardless of PrEP formulation (e.g., adaptive interventions [72]). For women who consistently use PrEP, adherence counseling support could be scaled down if desired. A future effectiveness trial and preliminary exploration of implementation strategies in collaboration with the Ugandan Ministry of Health will inform best approaches for public sector implementation and future scaling to meet the needs of women of reproductive age. In these next steps, we will evaluate the effectiveness and integration of this approach into routine care to improve PrEP uptake and adherence, including the cost, feasibility, scale-ability, and implementation process in Uganda.

Our screen to enroll ratio was high (916:131, 0.14) and may raise questions about the generalizability of the intervention. Over half of those excluded denied personal or partner pregnancy in the next year. Unfortunately, people who want to have children while exposed to HIV often face internalized, community, and healthcare stigmas around their choices given the risk of perinatal HIV acquisition [73–76]. Admitting being in an HIV-serodifferent partnership can also lead to stigma [77,78]. Thus, some eligible people may have screened out due to social desirability bias. In addition, most of our recruitment occurred through HIV testing and counseling sites. Further reach into the community and non-HIV focused clinical settings could enhance generalizability in future iterations [78]. Finally, many women do not plan for pregnancy and this may have led to limitations in recruitment. In future iterations of this work, we hope to adapt the intervention to also meet the HIV prevention needs of women with pregnancy and postpartum, thus expanding the reach and potential impact of the intervention. As opportunities to prevent HIV in the context of reproductive goals become more common, stigma may reduce.

Emerging studies of PrEP use among adolescent girls, and young women have mixed results, although most note high PrEP initiation and declining use over time [79–83]. The high uptake and sustained adherence observed in this study are consistent with a recent periconception study conducted among HIV−negative Kenyan women within a serodifferent partnership and desire to have a child [27]. In the Kenyan pilot study, PrEP was initiated by 100% of the HIV−negative partners (including 40 couples in which the woman was HIV−negative) and 81% took at least 80% of PrEP doses 1 month prior to pregnancy based on electronic pillcaps [27]. Another recent periconception study in the US observed that among 25 women taking PrEP while seeking conception with a male partner living with HIV, 87% had TFV-DP levels consistent with taking at least 4 doses/week [84]. Indeed, our qualitative sub-study with participants and partners suggested that the motivation to have a safe option to fulfill the cultural expectations of having children while maintaining serodifferent relationships encouraged PrEP adherence [78]. Conversely, a prospective cohort study in South Africa to assess the uptake and effectiveness of a safer conception intervention found lower PrEP initiation at 51% (22 of 43) among HIV−negative women in serodifferent or unknown serostatus relationships, but this study was conducted early in South Africa's PrEP roll-out [85]. Overall, our data and the literature suggest that the periconception time period, when women may be motivated to achieve reproductive goals and deliver an HIV-uninfected baby may help them to overcome oral PrEP adherence challenges [34].

More data have been collected from women using PrEP during pregnancy with conflicting results [31,32]. In an implementation project in Kenya, 22% of 9,376 pregnant and postpartum women and 79% of 193 pregnant and postpartum women with a partner known to be living with HIV initiated PrEP. Only 39% of these women continued to use PrEP after the first month [31]. In South Africa, in a cohort of 1,201 HIV–negative, pregnant women accessing care at a public sector clinic, 84% chose to use PrEP and 58% returned for 3-month follow-up [32]. Among those women, just 19% of pregnant women and 11% of postpartum women had drug concentrations consistent with taking 2 to 6 doses per week [86]. We observed ongoing high adherence, by pillcap, among women during pregnancy, suggesting that starting PrEP prior to pregnancy may provide an opportunity to overcome adherence barriers prior to onset of pregnancy symptoms or simply a selection bias that women who choose to use periconception PrEP are likely to have ongoing success. We have limited tenofovir concentration data from pregnant people in this study and will be conducting a future combined analysis with another dataset. While the periconception period is not routinely identified in general clinical care, we maintain that this may be a key opportunity to engage women in HIV prevention, including PrEP care.

Interpretation of our data varies to some extent by the adherence measure used and the degree to which a given adherence threshold correlates with PrEP effectiveness. We present various approaches to data interpretation to learn from the data, while recognizing these limitations. We categorized ≥80% pillcap adherence as high based on trends in the literature and associations with protection [61,62]. In vitro pharmacokinetic (PK) data suggest that 6 to 7 doses per week may be required to achieve stable FTC and TFV drug concentrations in cervical and vaginal mucosal tissues (86% to 100% of doses); however, which compartment requires what drug concentration to achieve protection from HIV remains unclear for receptive vaginal intercourse [62]. We categorized concentrations of ≥40 ng/mL for plasma TFV as high informed by data that includes African women, suggesting protection from this level, which reflects dosing in the past 24 h [20,63]. Whole blood concentrations of TFV-DP associated with protection from HIV for women are not available. Twenty women from East and Southern Africa who were 6 to 12 weeks postpartum and taking directly observed oral PrEP 7 times per week had 25th percentile concentrations of TFV-DP of 1,053 fmol/punch [67], and the corresponding estimate for 4 doses per week was 600 fmol/punch. For 25 US women (5 of whom were black, none pregnant), the 25th percentile concentration for steady state dosing at 4 doses per week was approximately 700 fmol/punch [66]; these data, plus data from men, has informed use of a concentration of 700 fmol/punch to indicate high adherence [64–67]. However, because data suggest that this cutoff is not sensitive for adherence in African cisgender women [68], we used 600 fmol/punch.

Differences observed across the PrEP adherence measures in this cohort highlight the value of incorporating multiple adherence measurements and expose some gaps in our understandings of tenofovir drug-level data interpretation for cisgender women. Importantly, adherence by any one of these measures was higher than seen in many prior cohorts of women of reproductive age. Further, 1 HIV seroconversion was observed despite high pregnancy and STI incidence indicating that the intervention supported women to avoid HIV transmission over time [87]. Exploring possible reasons for discrepancies across measures is important for interpreting the effect of our intervention and informing future analyses. First, the categories of high are not equivalent across the measures in terms of dosing or time period: 80% of pills taken by electronic pillcap equates to 5 to 6 doses per week averaged over approximately 90 days, ≥40 ng/mL represents dosing in the last 24 h, and ≥600 fmol/punch TFV-DP represents 4 or more doses per week over 6 to 8 weeks. In addition, the samples represent different groups: pillcap data were available for 83%, plasma data were processed for 39%, and whole blood samples

were processed for 93% of women accessing PrEP at the 3-month (primary outcome) time point. High adherence by electronic device persisted, whereas the proportion with high levels waned over time with plasma and whole blood measures. Women may have persisted using the pillcap device without taking pills due to social desirability bias [88]. However, studies have shown that adhering to pillcap opening without taking medication is difficult to maintain over time [89] and PrEP use was not required to remain in the study. Plasma TFV measures pill taking in the past week and women may have been prompted to take doses by reminders about upcoming clinic visits thus explaining higher plasma levels even when TFV-DP concentrations were lower; however, correlations between plasma TFV and electronic pillcap data suggest that pillcap use over 30 days aligned with an objective, biologic measure in the week prior to blood draw. Correlations between electronic pillcap and whole blood (DBS) were low. In a secondary analysis of whole blood spots collected from men and women with electronic pillcap data in the Partners PrEP demonstration project, TFV-DP measurements were specific but not sensitive for electronic pillcap adherence [68]. Those authors speculate that the discrepancy may relate to interindividual differences in drug disposition [90–92]. While interindividual differences in PK are certain, intraindividual PK changes over time are less likely. Thus, the downward trends of DBS TFV-DP and plasma TFV values may at least partially reflect behavioral fatigue with daily pill taking and highlight the importance of longitudinal adherence support for daily PrEP use. The most conservative estimates of adherence also suggest that some women may require different adherence support than that provided in this study; ongoing work is needed to optimize adherence support for women [72,93]. Again, adherence by all measures was higher in this cohort than for most PrEP implementation cohorts among women.

We did not identify factors that predicted PrEP use over time in our models. Covariates were selected based on our conceptual framework and further refined by our qualitative data and the literature of factors associated with PrEP use in other cohorts of women. We may have been under-powered to detect associations, particularly because pillcap adherence was consistently high for most women (confidence intervals for associations were wide.). In terms of uptake, we observed that women who initiated PrEP in this study had higher PrEP optimistic beliefs compared to those that did not initiate PrEP (based on average PrEP optimism score). Our qualitative data also suggested a high perception of PrEP effectiveness among participants [78]. This finding aligns with a recent study conducted in central Uganda showing high acceptability and willingness to take PrEP if offered among high-risk populations in Uganda [94]. In their study, which included a 2-day training workshop on PrEP for healthcare workers, investigators observed an increase in PrEP interest and knowledge 9 months post intervention.

In many trials of PrEP for women, male partner engagement is associated with higher adherence [95–98]. Due to the challenges with HIV-serostatus disclosure globally and the desire to offer HIV prevention to women who may not know her partner's serostatus or be able to engage him in HIV prevention, our study was designed to enroll women as individuals. The absence of required partner involvement does not appear to have dampened enthusiasm for PrEP use. From our qualitative data, some participants reported that the safer conception program encouraged them to disclose their serostatus with partners, support each other to ensure daily medication adherence, and offered a sense of hope to "fight" the virus together [78]. Given the limited prevention options for women who choose to conceive with men with HIV, WHO guidelines identify serodifferent couples considering conception as a priority group for PrEP [99]. The FDA labeling information and the US Perinatal ART Guidelines support periconception PrEP use [34,100–104]. Similarly, WHO and CDC guidelines emphasize

that eliminating perinatal transmission requires pre-conception counseling to reduce transmission to the mother and therefore the child [104,105].

Strengths of this study include the unique population, use of prospective data, use of well-validated tools for evaluating potential social and behavioral factors that could influence PrEP use, and objective measurements of PrEP adherence using daily electronic monitoring device and plasma and whole blood drug levels. However, this study also has limitations. First, due to intermittent countrywide stock-outs of HIV testing kits, attendance at HIV testing centers was low during much of the recruitment period, resulting in slower than expected recruitment. Second, while PrEP was being rolled out in this clinic in Uganda at the time of our study, women could access enhanced counseling and personalized pharmacy services in our program. Therefore, women may have enrolled in our program regardless of pregnancy plans but to access PrEP outside of the ART pharmacy programs where stigma may impair uptake. Finally, due to funding restrictions and study design, we have limited pregnancy data follow-up.

In conclusion, we found that PrEP is a desirable (high uptake) tool for HIV prevention among women vulnerable to HIV during periconception and pregnancy periods. These data suggest that women planning for and with pregnancy are able to use PrEP effectively and should be prioritized for PrEP implementation and adherence support, particularly in settings with high fertility rates and generalized HIV epidemics. Future work will aim to evaluate implementation of the Healthy Families-PrEP intervention on periconception, pregnancy, and postpartum PrEP use in public sector clinics in Uganda.

## Supporting information

**S1 Text. Table 1 scoring details.**
(DOCX)

**S2 Text. Full models for Table 2.**
(DOCX)

**S3 Text. Summary of plasma, DBS, Wisepill data at 3-, 6-, and 9-month visits.**
(DOCX)

**S1 STROBE checklist. STROBE checklist.**
(DOCX)

**S1 Protocol. Study protocol and analysis plan.**
(PDF)

## Acknowledgments

We gratefully acknowledge the contributions of the participants, families, and the entire study team for their contributions. This work is dedicated to the memory of our dear colleague, mentor, and friend, Dr. Mwebesa Bosco Bwana, who led our Healthy Families team and this study with deep compassion for the patients he loved to serve.

## Author Contributions

**Conceptualization:** Lynn T. Matthews, Christina Psaros, Craig W. Hendrix, David R. Bangsberg, Jessica E. Haberer, Mwebesa Bosco Bwana.

**Data curation:** Lynn T. Matthews, Kara Bennett.

**Formal analysis:** Lynn T. Matthews, Oluwaseyi O. Isehunwa, Kathleen E. Hurwitz, Kara Bennett.

**Funding acquisition:** Lynn T. Matthews, David R. Bangsberg.

**Investigation:** Esther C. Atukunda, Moran Owembabazi, Kato Paul Kalyebera, Christina Psaros, Pooja Chitneni, Mark A. Marzinke, Peter L. Anderson, Winnie Muyindike.

**Methodology:** Lynn T. Matthews, Craig W. Hendrix, Peter L. Anderson, Kathleen E. Hurwitz, Jessica E. Haberer.

**Project administration:** Esther C. Atukunda, Moran Owembabazi, Winnie Muyindike, Mwebesa Bosco Bwana.

**Supervision:** Lynn T. Matthews, Winnie Muyindike, David R. Bangsberg, Jeanne M. Marrazzo, Mwebesa Bosco Bwana.

**Writing – original draft:** Lynn T. Matthews, Oluwaseyi O. Isehunwa, Kara Bennett.

**Writing – review & editing:** Lynn T. Matthews, Oluwaseyi O. Isehunwa, Kathleen E. Hurwitz, Kara Bennett, Jessica E. Haberer.

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
