## [Editor Report · Decision Letter 0]

8 Aug 2022

Dear Dr Matthews, 

Thank you for submitting your manuscript entitled "High PrEP uptake and objective longitudinal adherence among HIV-exposed women with personal or partner plans for pregnancy in rural Uganda" for consideration by PLOS Medicine.

Your manuscript has now been evaluated by the PLOS Medicine editorial staff and I am writing to let you know that we would like to send your submission out for external peer review.

Please re-submit your manuscript within two working days, i.e. by Aug 10 2022 11:59PM.

Kind regards,

Beryne Odeny, PhD

PLOS Medicine

---

## [Decision Letter · Decision Letter 1]

31 Oct 2022

Dear Dr. Matthews,

Thank you very much for submitting your manuscript "High PrEP uptake and objective longitudinal adherence among HIV-exposed women with personal or partner plans for pregnancy in rural Uganda" (PMEDICINE-D-22-02648R1) for consideration at PLOS Medicine. 

Your paper was evaluated by an associate editor and discussed among all the editors here. It was also discussed with an academic editor with relevant expertise, and sent to independent reviewers, including a statistical reviewer. The reviews are appended at the bottom of this email and any accompanying reviewer attachments can be seen via the link below:

[LINK]

In light of these reviews, I am afraid that we will not be able to accept the manuscript for publication in the journal in its current form, but we would like to consider a revised version that addresses the reviewers' and editors' comments. Obviously we cannot make any decision about publication until we have seen the revised manuscript and your response, and we plan to seek re-review by one or more of the reviewers. 

We hope to receive your revised manuscript by Nov 21 2022 11:59PM. Please email us (plosmedicine@plos.org) if you have any questions or concerns.

We look forward to receiving your revised manuscript. 

Sincerely,

Callam Davidson, 

PLOS Medicine

plosmedicine.org

Comments from the Academic Editor:

In particular, I would like to see the authors respond to:

Reviewer 1 point 8 (representativeness/generalizability)

Reviewer 2 point 4 and Reviewer 3 (transmission despite high adherence)

Reviewer 4 point 1 (feasibility) and point 2 (measure) and all comments on Discussion.

Expanding on one of Reviewer 4's comments, I'd also like to see a much more thorough discussion of the feasibility of scaling up this intervention. Making PrEP available is clearly not sufficient to achieve both high uptake and good adherence. The counseling and support (and perhaps frequent monitoring) that the authors offered must have mattered, but I saw no discussion of this or how such services might be implemented at large scale in routine care. It's not clear whether it would easy or difficult to do so, or what resources are required. I also wonder if the regular study monitoring improved PrEP adherence.

Finally, Reviewer 1's point about the representativeness of this population seems important. The large proportion of potential participants who screened out because they did not want a baby (more than half those screened, Figure 2) seems a bit at odds with the claims about high fertility in this population and makes me wonder if targeting of this intervention is needed.

Please update your title to ‘High PrEP uptake and objective longitudinal adherence among HIV-exposed women with personal or partner plans for pregnancy in rural Uganda: a cohort study’

Please structure your abstract using the PLOS Medicine headings (Background, Methods and Findings, Conclusions).

Abstract Background: The final sentence should clearly state the study question.

Abstract Methods and Findings:

* Please ensure that all numbers presented in the abstract are present and identical to numbers presented in the main manuscript text.

* Please include the important dependent variables that are adjusted for in the analyses.

Please confirm whether author KEW’s affiliation with NoviSci poses a potential competing interest that ought to be declared.

The author contributions section can be removed (this will be captured as metadata via the Submission Form), and the Acknowledgements section can be relocated to the end of the manuscript.

Please include continuous line numbering throughout the manuscript.

Please place citations in square brackets throughout.

Please ensure that the study is reported according to the STROBE guideline, and include the completed STROBE checklist as Supporting Information. Please add the following statement, or similar, to the Methods: "This study is reported as per the Strengthening the Reporting of Observational Studies in Epidemiology (STROBE) guideline (S1 Checklist)."

Did your study have a prospective protocol or analysis plan? Please state this (either way) early in the Methods section.

Please cite specific items located in the Supporting Information as outlined here (as opposed to stating ‘Appendix’ only): https://journals.plos.org/plosmedicine/s/supporting-information

Please specify whether informed consent was written or oral.

Please place each table in your manuscript file directly after the paragraph in which it is first cited (read order). Do not submit your tables in separate files.

‘Periconception PrEP Adherence’ and ‘Periconception Drug Concentrations’: PLOS does not permit data not shown – please include the relevant data (in the Supporting Information if necessary). 

Some references are lacking information (e.g., 12-15 and 101-102 should contain URLs and dates of citation where appropriate). See https://journals.plos.org/plosmedicine/s/submission-guidelines#loc-references for more information.

Reference 32 is a preprint – please confirm it has not yet been published and, if not, please add [preprint] to the reference after ‘medRxiv’.

Comments from the reviewers:

Reviewer #1: Statistical review

This paper reports an observational study of an intervention to encourage PrEP initiation amongst women planning pregnancy in Uganda. The paper shows high adherence to the intervention. The statistical methods used are appropriate, although some extra detail is recommended (see below). I have some comments, which I have listed below.

1. Title: should mention the study design (i.e. prospective cohort study)

2. Abstract: it's not too clear from the abstract what the main study objectives were and how these link to the results: referring to the healthy families intervention but not to the effect of the intervention made the abstract a bit confusing (in isolation) to me.

3. Methods: as per above comment it would be good to have a (small) subsection that describes the main objectives of the study. 

4. Methods: It would be good to have some text that describes how the sample size was chosen. This may not have been chosen on statistical grounds, perhaps on feasibility grounds.

5. Methods, statistical analysis: it would be useful to add whether the statistical analysis was pre-specified and, if so, provide a statistical analysis plan.

6. Methods, statistical analysis: I found the model used for secondary adherence analysis hard to visualise - could more be added in the appendix on this and how to interpret the parameters?

7. Results: I believe PLOS medicine does not permit 'data not shown' so these results could be provided in supplementary material. I think the adherence data are useful to provide as relevant to other research studies.

8. Discussion: it would be useful to discuss how representative the participants who enrolled in the study were of the broader target population. Are results from this study likely to be relevant in the broader population?

James Wason

Reviewer #2: High PrEP uptake and objective longitudinal adherence among HIV-exposed women with personal or partner plans for pregnancy in rural Uganda 

Thank you for allowing me to look at this manuscript. It has very important results useful to low resource-settings with high HIV prevalence and high fertility settings.

Comments:

1. Page 12: The ranges overlap. If the participant of 4 doses:

TFV-DP reflects average dosing in the last 6-8 weeks; >600 fmol/punch indicates approximately >4 doses per week (categorized as high adherence), 450-600 fmol/punch indicates about 3- 4 doses per week (categorized as moderate adherence), and bet 

2. Pregnancy incidence: Please explain how you get the pregnancy incidence of 75% at (53 pregnancies of 131 women) 

A total of 53 pregnancies occurred among 131 women with 848 person-months of follow-up (median 8.8 months), resulting in a pregnancy incidence of 75% (95% CI: 57%, 98%) (Figure 2). 

3. Page 15: Periconception Drug Concentrations. 

All processed samples were from women who were not pregnant at the time of collection. 

Please explain why you used only not pregnant participants for this sample collection. Could it be different than that from those who got pregnant?

4. I expected some explanations of this in the discussion:

One participant tested positive for HIV at 9 months. This participant was not pregnant. Although electronic adherence data suggested high PrEP adherence (>80% of doses all months), plasma TFV levels were undetectable at 3, 6, and 9 months of follow-up. 

5. Typos. Need to read through the document and correct the typographic errors

e.g ….. counselors (social desirability bias)(85). However, studies have shown that adhering to pill cap opening without taking medication is difficult to maintain over time(86); further, PrEP use was not required to remain in the study. 

Reviewer #3: I commend the authors for this manuscript that evaluates an are of great public health importance. The study explores important application of HIV prevention intervention in reproductive health programs. The manuscript is well written and has used vigorous analysis approaches to evaluate uptake and adherence of PrEP in the periconception period. The numbers are relatively small and thus denies the investigators ability to draw some conclusions in some scenarios where they acknowledge being underpowered. This may be the major shortcoming of this study, however this is not fatal to their work.

I was surprised with the findings on the participant who seroconverted in the course of the study. This participant had high adherence by electronic adherence data however, her plasma levels were undetectable at month 3, 6 and 9. This, coupled with the results showing poor correlation between TFV-DP and electronic adherence data suggests caution in interpretation of results involving electronic adherence data. I think this may be a contribution of this work in understanding adherence. Would the authors consider firming up their position on this in their manuscript? They appear to remain non-committal.

Finally, in the discussion, on a paragraph that appears with ref #45 and #79 rectify this typo error

"to predict which women" and NOT "to predict who which women"

Reviewer #4: This is an interesting manuscript about an important priority population for HIV prevention - women with periconception risk of HIV acquisition. There is discussion of an exciting new intervention as well as very robust measurement of adherence to PrEP using multiple measures and the authors should be congratulated on the work. Overall, the adherence to PrEP in this study was impressively high compared to other recent cohorts and deserves attention by the field. However, I do think that the overall focus of the manuscript could be clarified and strengthened to make this paper more accessible to readers. I hope the comments below are helpful to the authors in revising the manuscript.

Major comments:

1) In terms of the overarching focus of the manuscript it was somewhat unclear to me whether the focus was on the development and evaluation of impact of a novel intervention or on a detailed comparison of adherence metrics within the cohort. The abstract (Methods) states women were enrolled to evaluate impact of intervention but there is no comparator group against which to evaluate. Each of these questions (intervention and adherence metrics) is interesting and it would have been helpful to more clearly describe/separate in abstracts/results/discussion. I was also interested in more data on how the intervention was delivered/by whom, how long it took, acceptability of sessions - if this is to be evaluated further and potentially scaled up. If the focus is on the intervention, would suggest including figure of conceptual model in main paper if possible or in supplemental figures, and also including more on the intervention in the Discussion

2) Regarding the adherence data, the adherence via objective metrics was quite high in this study compared to other studies in similar populations/settings. However, given that multiple studies have suggested that electronic pillbox openings overestimate adherence and (as authors note) plasma reflects only very recent adherence - would suggest that the authors focus on the TFV-DP metrics as a cumulative metric more reflective of adherence over time in the study (recognizing that not all participants had DBS measures). Even using the TFV-DP metric, about 45% with high adherence and 60% with any adherence at month 9 is still much higher than in many recent studies - although this was not a primary outcome would suggest emphasizing levels of adherence in study around these #s rather than 80% which is likely an overestimate. It is good that the authors acknowledge some uncertainty around appropriate cutoffs for TFV-DP in DBS for African women and would suggest bringing this point in earlier when the thresholds are described in the methods. 

Additional comments:

ABSTRACT

1) Could the authors please briefly describe setting from which participants were recruited and where visits occurred (e.g. general ANC clinic/hospital/community)?

2) Methods - consider rewording "impact" of intervention given no comparator (pre/post or control arm)

METHODS

1) Population and eligibility (top of page 9) - how was self-assessed risk determined

2) Healthy living intervention Figure 1- text at bottom of figure states 12 weeks from enrollment to completion of all sessions but box on Sessions 2,3 states delivered every 3 months - could the authors clarify how often sessions were delivered and total follow-up time to deliver the intervention?

3) Study procedures - participants received a package of HIV prevention OR safer conception counseling - were both offered or only one (should this be AND?)

4) Middle of page 10 - baseline HIV testing - could the authors kindly confirm if this was POC 4th gen Ag/Ab or rapid 3rd gen Ab? To my knowledge 3rd gen is usually SOC in this setting rather than POC 4th gen but would appreciate clarificiation.

5) Middle of page 12 - discussion of thresholds for TFV-DP - would be helpful to mention here that the 600 fmol/punch differs from prior thresholds in earlier studies e.g. MSM and how 600 was selected

6) Statistical analysis - for the adjusted model - was adjustment based on the initial causal model or stepwise removal? 

RESULTS

1) Periconception drug concentrations - were any blood samples collected from women who were pregnant? Is figure 4 only showing data from women who were not pregnant, e.g. at 9 months of follow-up? I understand initially women were exited from study at time pregnancy identified but adherence during pregnancy would also be of interest and expected to see some data here - thank you for insights/clarification

2) Middle of page 15 - on drug levels - authors state comparison of electronic and plasma data .. "suggesting that pill taking behavior over longer time frames correlates well with objective adherence behavior reflecting pill taking over the past week" - would consider revising this sentence given the subsequent DBS data and that electronic pillbox openings does not necessarily mean pill ingestion. As noted later in paragraph, electronic data did not correlate with TFV-DP levels.

3) Figure 3b seems to come after Figure 4 - possible to renumber for clarity? 

DISCUSSION

1) Consider emphasizing (as in half-way through paragraph 1) that about half of women consistently took oral PrEP based on TFV-DP - agree that this is still quite a high percentage

2) In Paragraphs 2 and 3 would have been interested in more emphasis on the results of this study rather than results from other studies

3) Paragraph 4 on intervention - would have been interested (as noted above) in more results and discussion of this novel intervention

4) Conclusion - I agree with all of the statements in general, e.g. women planning for pregnancy should be prioritized for PrEP - but would appreciate more clarity on how the results of this study lead to the conclusion.

[LINK]

---

## [Decision Letter · Decision Letter 2]

10 Jan 2023

Dear Dr. Matthews,

Thank you very much for re-submitting your manuscript "High PrEP uptake and objective longitudinal adherence among HIV-exposed women with personal or partner plans for pregnancy in rural Uganda: a cohort study" (PMEDICINE-D-22-02648R2) for review by PLOS Medicine.

I have discussed the paper with my colleagues and the academic editor and it was also seen again by two reviewers. I am pleased to say that provided the remaining editorial and production issues are dealt with we are planning to accept the paper for publication in the journal.

[LINK]

We look forward to receiving the revised manuscript by Jan 17 2023 11:59PM.   

Sincerely,

Callam Davidson, 

Senior Editor 

PLOS Medicine

plosmedicine.org

Requests from Editors:

You note in the Acknowledgements that Gilead Sciences reviewed the final manuscript – please include this information in your Financial Disclosure and Competing Interests. Additionally, please indicate whether Gilead Sciences played any other role in the study design, data collection and analysis, decision to publish, or preparation of the manuscript?

Please structure your Abstract using PLOS Medicine headings: Introduction, Methods and Findings, Conclusions (the Methods and Findings sections ought to be combined). 

Please relocate the sentence ‘Study design limitations include lack of a control group’ to the end of the Methods and Findings section (final sentence before Conclusions section).

Please include continuous line numbering throughout the manuscript (in the last revision this only appeared in the .doc version and not the PDF).

The Disclosures and Funding sections can be removed from the main text, please ensure all details are captured in the relevant answer from the submission form (Competing Interests). 

Please format your Author Summary using 2-3 single sentence bullet points per question.

Please provide legends for all Figures (including those in the Supporting Information).

Please define ‘lost to follow up’ as used in this study.

Table 2: Please include the covariates adjusted for in the legend.

Please enlarge the text in Figure 4 as it is currently difficult to read. 

In the interest of brevity and clarity, please trim the additional Discussion text relating to transmission despite high adherence.

References: Please include date cited for any internet sources.

Please delete the competing interests information from references 28 and 61.

Comments from Reviewers:

Reviewer #1: Thank you to the authors for addressing my previous comments well. I have no further issues to raise.

Reviewer #4: The authors were highly responsive to the reviews. I appreciate the opportunity to re-review the manuscript and the additional details on the intervention are very helpful. I recommend acceptance with if one suggested change as follows in the Abstract - 

Abstract

The objective has been changed from "We conducted a longitudinal cohort study in Uganda to evaluate oral PrEP uptake and adherence as part of HIV prevention in the context of reproductive goals for women (i.e., safer conception care)." to "We conducted a longitudinal cohort study to evaluate whether Healthy-Families-PrEP supported sufficient oral PrEP use to reduce HIV incidence among women."

The revised sentence including the phrase "to reduce HIV incidence" is somewhat misleading because HIV incidence is not the primary outcome of the study and the sample size is rather small for incidence measurement. Would suggest that the authors reword the sentence to focus on uptake and adherence rather than incidence.

Otherwise, recommend acceptance.

[LINK]

---

## [Editor Report · Decision Letter 3]

23 Jan 2023

Dear Dr Matthews, 

On behalf of my colleagues and the Academic Editor, Professor Sydney Rosen, I am pleased to inform you that we have agreed to publish your manuscript "High PrEP uptake and objective longitudinal adherence among HIV-exposed women with personal or partner plans for pregnancy in rural Uganda: a cohort study" (PMEDICINE-D-22-02648R3) in PLOS Medicine.

PRESS

Sincerely, 

Callam Davidson 

Associate Editor 

PLOS Medicine